# Advanced Deep Learning Models for Melanoma Diagnosis in Computer-Aided Skin Cancer Detection

**DOI:** 10.3390/s25030594

**Published:** 2025-01-21

**Authors:** Ranpreet Kaur, Hamid GholamHosseini, Maria Lindén

**Affiliations:** 1Department of Software Engineering & AI, Media Design School, Auckland 1010, New Zealand; 2School of Engineering, Computer, and Mathematical Sciences, Auckland University of Technology, Auckland 1010, New Zealand; hamid.gholamhosseini@aut.ac.nz; 3Division of Intelligent Future Technologies, Mälardalen University, 721 23 Västerås, Sweden; maria.linden@mdh.se

**Keywords:** skin cancer, melanoma, classification, segmentation, deep learning

## Abstract

The most deadly type of skin cancer is melanoma. A visual examination does not provide an accurate diagnosis of melanoma during its early to middle stages. Therefore, an automated model could be developed that assists with early skin cancer detection. It is possible to limit the severity of melanoma by detecting it early and treating it promptly. This study aims to develop efficient approaches for various phases of melanoma computer-aided diagnosis (CAD), such as preprocessing, segmentation, and classification. The first step of the CAD pipeline includes the proposed hybrid method, which uses morphological operations and context aggregation-based deep neural networks to remove hairlines and improve poor contrast in dermoscopic skin cancer images. An image segmentation network based on deep learning is then used to extract lesion regions for detailed analysis and calculate the optimized classification features. Lastly, a deep neural network is used to distinguish melanoma from benign lesions. The proposed approaches use a benchmark dataset named International Skin Imaging Collaboration (ISIC) 2020. In this work, two forms of evaluations are performed with the classification model. The first experiment involves the incorporation of the results from the preprocessing and segmentation stages into the classification model. The second experiment involves the evaluation of the classifier without employing these stages i.e., using raw images. From the study results, it can be concluded that a classification model using segmented and cleaned images contributes more to achieving an accurate classification rate of 93.40% with a 1.3 s test time on a single image.

## 1. Introduction

Melanocytes proliferate, and fibrous tissues overgrow in the deeper layers of the skin, thereby causing skin cancer. In the outermost layer of the skin, cancerous cells are visible. Several factors contribute to the development of skin cancer, including fair complexion, overexposure to sun rays, sunburn, genetic history, and weak immunity [1,2]. In many cases, skin cancer is indistinguishable from a normal mole due to its shape, size, and color. As a result, it is challenging to spot with the naked eye. There are main five types of skin cancer [3], as shown in Figure 1. It is crucial to be able to distinguish melanoma (MEL) from other types of skin cancer such as basal cell carcinoma (BCC), benign keratosis-like (BKL) lesions, congenital melanocytic nevus (CMN), and benign (BEN). Since MEL is an aggressive form of skin cancer, it is less common than the other types. The other four types of cancer are considered benign because they do not invade other parts of the body. These cancers do not spread to local structures or distant parts of the body. The growth of benign tumors is usually slow, and their borders are clearly defined. The majority of benign tumors are not dangerous. Hence, the primary objective of this study was to detect melanoma and distinguish it from benign cancer.

Image-assessment-based diagnosis is crucial in diagnosing abnormalities, including skin cancer, all over the body. The American Academy of Dermatologists Association (AADA) [4] estimates that one out of every five Americans develops skin cancer at some point in their lives. A total of 197,700 new cases of melanoma occurred in the U.S. in 2022, including 97,920 noninvasive (in situ) cases and 99,780 invasive cases [5,6]. The International Agency for Research on Cancer (IARC) also pointed out in its latest study [7] that there will be an increase in cutaneous melanoma cases by more than 50% from 2020 to 2040. In Australia and New Zealand, the incidences observed for melanoma are 42 cases per 100,000 men and 31 cases per 100,000 women, followed by Western Europe (19 cases in both sexes), Northern America (18 cases in men and 14 in women), and Northern Europe (17 cases in men and 18 cases in women) [7]. The incidence of melanoma is low in most countries in Africa and Asia, where it is commonly less than 1 per 100,000. By 2040, scientists estimated that nearly 100,000 melanoma deaths worldwide will have occurred due to global population changes, and more than 500,000 new cases of melanoma are expected every year.

The early detection of melanoma is imperative for ensuring high survival rates. If diagnosed early, the disease can be successfully treated. A CAD system can diagnose a wide range of disorders quickly, efficiently, and consistently. In addition to offering precise and cost-effective disease detection and protection, CAD also offers advanced tumor disease detection and protection. A variety of imaging technologies are used to assess disorders of the human organ system, including magnetic resonance imaging (MRI), positron emission tomography (PET), and X-rays. For skin lesion analysis and prognosis, dermatoscopes are used to visually examine the infected skin area with a magnified view.

In the current clinical setting, however, there are a few limitations that complicate diagnosis. A dermatologist, for instance, examines a skin lesion visually based on various characteristics such as its size, border, shape, and color, called the ABCDE criteria or seven-point checklist [8]. There are, however, inaccuracies due to complex lesion patterns and variations in interpretation. A second issue is that skin cancer diagnosis is an expensive procedure, as the average treatment costs in the USA for existing patients are approximately USD 44.9 million, and for newly diagnosed patients, it is approximately USD 932.5 million [9]. In addition, the accuracy of skin lesion diagnostics is lower among dermatologists with little expertise. Physicians use a variety of subjective, time-consuming, and error-prone methods to evaluate and analyze lesion images. Because skin lesions are so complex, the images are difficult to understand. Analyzing images for skin lesions, evaluating them, and being aware of them requires unambiguous identification of lesion pixels. Thus, computer-aided diagnostic and prediction systems for skin cancer detection have made significant advances by utilizing deep learning approaches in computer vision. These computer-aided models are affordable and fast to analyze initial skin lesions.

The purpose of this study was to develop a variety of deep learning models for the different phases of a computer-aided melanoma diagnosis system, which utilizes lesion image sites to correctly diagnose the early stages of cancer. The system under consideration relies on preprocessing, segmentation, and classification operations. The major novel contributions made to each phase such as noise removal, image contrast enhancement, segmentation, and classification of the melanoma CAD system in this research are listed as follows:Reconstruction of the entire input image using a custom-designed intensity-adjustment-based interpolation technique after eliminating hairlines (Section 3.2.1).Application of a context aggregation deep learning network to improve image contrast, as well as rescaling and standardization of images to fit the training models during the learning procedure (Section 3.2.2).Design of a novel deep neural network incorporating atrous dilated convolutions to extract lesion segmentation maps with high spatial resolution (Section 3.3).Design of a novel deep convolutional neural network (N-DCNN) to calculate the high- to low-level features of a lesion to distinguish benign and malignant lesions (Section 3.4).Performance analysis of the classification model utilizing unprocessed (raw images) and preprocessed (enhanced images) datasets.

## 2. Related Studies

Several categorization algorithms and strategies have been devised to enhance the accuracy of the melanoma diagnosis process. This study focused on deep learning models for different stages of skin cancer in CAD systems to develop an optimized approach in contrast to a variety of other existing image classification models. In the case of skin cancer problems, the presence of hairlines is a major challenge because such kind of noisy data produce poor feature sets, which can further lead to poor image segmentation and classification results. The hairline removal method is a two-phase process: First, the localization and occlusion of hair pixels are performed using morphological operations [10,11], edge operators [12], and Gaussian derivatives [13]. Second, to repair the color and texture of occluded areas, bi-linear interpolation and histogram equalization methods [14] are used. There are old and limited studies [15,16] that contributed to the employment of hairline removal preprocessing methods to clean dermoscopic skin cancer images. The major problem not addressed in the literature is the detection and painting of very thin and complex hairlines in the skin samples used in the preprocessing phase of melanoma CAD models. Furthermore, the original color contrast and texture information of the image that has been lost after the hair pixel’s occlusion are not restored by the existing methods.

For the segmentation stage of a melanoma CAD model, several methods have been presented, such as machine-learning-based unsupervised [17], and optimization [18] methods. Due to the ineffectiveness of such approaches in producing accurate lesion segmentation masks, recent studies have utilized a variety of deep learning frameworks to extract information about lesions. CNNs, such as fully convolutional neural networks (FCNs) [19] and deconvolution networks [20], as extensions of FCNs, SegNet [21], and UNet [22], have been proposed by researchers as potential segmentation methods. These networks have some limitations, such as pooling layers that reduce resolution and discard essential image information. To perform semantic segmentation, class maps must be precisely known, and ‘where’ information must be preserved. For some problem domains, encoder–decoder networks (SegNet and UNet) maintain output image quality efficiently; however, their complexity and long execution times can cause them to be heavy. The concept of atrous convolutions, introduced by Chen [23], enables the direct control of the resolution to preserve the feature map information computed in the deep convolutional layers at the expense of long computation time due to their large networks. Therefore, there is a high demand for an optimized and less complex architecture that can accurately extract lesion information.

Further, for the last stage of the melanoma CAD model, several image-based skin classification systems have been developed using traditional machine learning techniques, such as K-nearest neighbors (KNN) [24,25,26], support vector machine (SVM) [26,27,28], artificial neural network (ANN) [29,30,31], and deep learning approaches [32]. Review articles on these traditional classifiers can be found [33,34]. The reviews point out that traditional machine-learning techniques are feature-dependent. The wrong selection of features can lead to poor classification results. Several deep learning techniques have been proposed for automated classification tasks [35,36,37]. For example, in 2012, Krizhevsky et al. [38] proposed AlexNet to classify 1.2 million images in the ImageNet containing 1000 different classes. A few attempts were also made by integrating traditional machine learning algorithms and deep convolutional neural networks (CNN) to improve the performance on classification problems. Ulzii et al. [39] proposed a technique that employs a pretrained AlexNet and the SVM classifier to classify 3753 skin lesion images. They explained the advantages of AlexNet in terms of its few training parameters and validity. Mahbod et al. [40] presented three deep learning networks, AlexNet, VGG16, and ResNet18, as feature generators. They trained the SVM classifier with feature sets from three neural networks and fused their results to make final predictions to detect three categories of skin cancer. They concluded in their study that fusing multiple features from various networks leads to a better classification rate. However, these approaches still require the additional step of crafting the feature sets employed in machine learning classifiers.

Some recent works have used optimized deep learning frameworks to enhance the melanoma classification performance. Guoda et al. [41] considered preprocessing and classification as key steps in achieving high performance. To reduce the inconsistencies among images, the entire intensity of the image was improved during the preprocessing stage. Then, a crossbred deep learning model was developed to predict cancer, and a comparison was made with other pretrained networks such as ResNet50, InceptionV3, and ResNet-Inception. Another study presented in [42] proposed a three-step superpixel method, which was further enhanced by mapping it through a new activation function. The deep feature extraction was performed via a pretrained ResNet50 deep learning model, and then features were optimized using a modified grasshopper method. However, that study did not consider the removal of hairlines and the impact of using segmented images instead of original images.

The literature suggests there is an increasing need for a high-performing skin cancer diagnosis system. To achieve high diagnosis accuracy, this research aimed to develop an automated classification system to determine malignant skin cancer. The above-discussed studies on skin cancer classification have utilized well-known deep learning architectures for classification purposes. However, deep learning frameworks for classification and segmentation still need to be enhanced and optimized as well as made more efficient. Moreover, to the best of our knowledge, the impact of employing results from the preprocessing stages on the final classification outcome was not examined in existing studies. This study also considered a complex and challenging skin cancer dataset from the ISIC 2020 challenge, which consists of several samples from dark-skinned people, the older population, and highly noisy conditions.

## 3. Materials and Methods

This research classified skin cancer into two types, benign and malignant, where benign further includes keratosis, basal cell carcinoma, and nevus lesions. Deep learning approaches were used for different phases of the melanoma CAD system for the classification of skin cancer. In Figure 2, a systematic methodology is presented that accepts the skin cancer image dataset, and its learning is performed in two ways. Step 1 performs classification of the raw images with a few basic preprocessing operations, and step 2 involves a series of phases such as preprocessing, segmentation, and classification. During the preprocessing phase, hairlines are obliterated using a hairline removal method, followed by a context aggregation deep learning model to regain the contrast and smoothness lost during hair detection and repatriation. In the segmentation operation, lesions are separated from the background using a neural network based on dilated atrous convolutions. In the last stage, melanoma and benign tumors are classified using an N-DCNN model. As a final step, performance metrics were used to compare and analyze the classification results of steps 1 and 2.

### 3.1. Dataset Preparation

In this study, a dermoscopic image dataset from ISIC 2020 [43,44] was used for training and testing the network. The ISIC is a well-known, open-access repository of dermatology images that presents a variety of challenging and complex skin cancer samples. There are 33,126 lesion samples in the original ISIC 2020 dataset collected from 2056 patients. The lesions were collected from men and women. Skin cancer is more common in men than women. In the given dataset, 15,981 lesions are from women, and 17,145 belong to men. In the dataset, 16 samples were collected for each patient to observe changes over time. Because skin cancer develops over time, its ABCDE (asymmetry, border, color, diameter, and evolution) characteristics change over time. Several of the lesions likely came from the same patient who developed symptoms of cancer, which is why their mole changed in shape, color, and size. For some patients, no changes were observed at the time of the next scan; thus, such images formed duplicates [45]. This may result in biases and overfitting in the network. To prevent these issues, data normalization was performed to eliminate duplicates. After removing duplicates, the dataset consisted of 12,882 skin samples, 7093 of which were BEN samples, and 5789 of which were MEL samples. There are also some challenging samples in the present dataset, including samples from people with dark skin and samples from older individuals. The dataset contains highly unbalanced classes, which may result in low accuracy of the model, for example, with significantly fewer MEL samples than BEN samples. Hence, synthetic samples were generated for classes with fewer images using random oversampling. Of the samples, 70% were used for training, 10% for validation, and 20% for testing.

### 3.2. Preprocessing

#### 3.2.1. Hairlines Removal Method

Various external and environmental factors can cause noise artifacts in skin cancer images. The main concern is the presence of hairlines and poor contrast, which compromise the model’s ability to diagnose. The proposed hairlines removal approach first transforms the color space from RGB to grayscale, and then hairlines are detected through morphological closing and subtraction operations.(1)$$\begin{array}{c} {\mathrm{Close}}_{\mathrm{img}}=\left(I\left(x,y\right)⊕\mathrm{SE}\right)⊖\mathrm{SE} \end{array}$$(2)$$\begin{array}{c} {\mathrm{Diff}}_{\mathrm{img}}={\mathrm{Close}}_{\mathrm{img}}-I \end{array}$$ The closing operation is performed between the original image *I* and the structuring element SE by applying dilation (⊕), followed by an erosion operation (⊖), as given in Equation (Equation 1). An image-closing operation removes the irrelevant parts of an image. In addition, it eliminates small holes, fills gaps between boundary lines, and smooths contours from the outside. To detect differences between two images and level uneven parts, a subtraction is performed between an image obtained after closing $Close_{img}$ and an original image *I*, as presented in Equation (2). After subtraction, the resultant image contains the pixels that differ between the two. A binarization process using T = 25 as a threshold is used to produce hair masks and refine hair pixels. During this comparison, background pixels are set to ‘0’, and foreground pixels, including those associated with hair, are set to ‘1’. To repair the boundaries and bridge the gaps in the resultant image, a dilation operation is applied.(3)$$\begin{array}{c} {\mathrm{Dilation}}_{\mathrm{img}}={\mathrm{Diff}}_{\mathrm{img}}>T \end{array}$$(4)$$\begin{array}{c} {\mathrm{Dilation}}_{\mathrm{img}}={\mathrm{Dilation}}_{\mathrm{img}}⊕\mathrm{SE} \end{array}$$(5)$$\begin{array}{c} {\mathrm{New}}_{\mathrm{img}}=\left\{\begin{array}{c} I,\;\mathrm{if}\;{\mathrm{Dilation}}_{\mathrm{img}}==0\;otherwise,\\ I_{\mathrm{new}},\;\mathrm{with}\;\mathrm{new}\;\mathrm{intensity}\;\mathrm{remapping} \end{array}\right. \end{array}$$ Furthermore, the pixels in a dilation image are copied into a new image based on a comparison: if a pixel value is ‘0’, then it is considered the background of the new image. The pixels having non-zero values are compared against the threshold to check if it is a hairy pixel, and, if detected, it is replaced with the intensity of the neighboring pixels. The above-given operations are explained in Equations (Equation 3)–(5). The image output produced after each operation of the hairline removal method is displayed in Figure 3.

#### 3.2.2. Contrast Enhancement Model

During the previous step of hair removal, the contrast and resolution of the skin images are significantly affected because the output image’s pixel was reconstructed, so it is necessary to apply a postprocessing method to enhance image quality. A deep-learning-based multiscale context aggregation network (MCAN) is used to approximate the contextual characteristics of skin cancer images. Detailed information about how hairlines are occluded and how MCAN improves medical image contrast can be found in our other research paper [46]. To achieve the highest accuracy, MCAN must first be trained on a set of images with high-frequency details. As a result of training, the network can process degraded images directly without the need for conventional processing. The network aggregates multiscale information of images and preserves local and global information at the deepest level of the network.

An illustration of the MCAN model’s architecture is shown in Figure 4. In this network, original medical images are extracted and divided into patches of 256×256 each. The MRABs (multiscale residual aggregation blocks) are the building blocks of the network. There are eight context modules in the MCAN, called MRAB, which consist of layers such as the input, 2D convolution, batch normalization, leaky ReLU activation, and regression layers. Multiscale contextual information is gathered by these context modules and passed on to the next MRAB module.

An image patch with a different dilation rate is applied with 32 convolutions in the MRAB. Each convolution layer $Conv_{i}$ …, $Conv_{N}$ has its own dilation rate set at 2,4,6,8,16,32,64, and 128, where N=8. To aggregate contextual information, the network uses dilation convolutions systematically without deteriorating the image’s quality. To ensure the normalization and identity branches are adjusted and strengthened, two more custom layers are created, adaptive normalization μ and λ. An initial training phase was conducted on original skin cancer images to estimate contextual information about the cancer images. Then, the model was applied to occluded hair images to restore their contrast.

### 3.3. Lesion Segmentation Model

Segmentation of lesions is a challenging task in CAD since it involves identifying the precise boundary of the lesions. For clinicians to better understand lesion patterns, an atrous convolutional neural network (ACNN) is used to segment infected areas from skin photographs automatically. The proposed segmentation deep learning model utilizes Chen et al.’s [23,47] original concept of atrous rates in the convolution operation. By incorporating atrous rates into the ACNN architecture as shown in Figure 5, a wide view of fields is achieved. The structure of ACNN is designed into 5 blocks having 16 feature extraction layers with different dilation factors. There are two primary steps in ACNN: pixel classification, or segmentation, and feature extraction. To enhance the filter’s view, the network uses multiple scaling rates combined with atrous convolutions. Segmentation models use enhanced input images through preprocessing methods such as hair removal and context aggregation, which are converted to a specified size and then processed by various sub-blocks to compute features.

In the network, the convolutional layer performs the feature calculations. An image $I_{x,y,k}$ is converted into a feature map FM(x,y,k) by sliding a small matrix called a kernel or filter over it and transforming the pixels’ values as follows:(6)$$\begin{array}{c} \mathrm{FM}\left(x,y,k\right)=\sum_{i=1}^{m_{h}}\sum_{j=1}^{m_{w}}\sum_{k=1}^{m_{c}}I_{x+i,y+j,k}*K_{i,j,k}\;\;r=1 \end{array}$$ The height, width, and number of channels of an input image $I_{[}x,y,k]$ can be defined as $m_{[}h]$, $m_{[}w]$, and $m_{[}c]$. The channels in an input image $I_{[}x,y,k]$ and kernel $K_{[}i,j,k]$ need to match. Each convolution layer was constructed using atrous convolutions with varying dilation rates. The filter can assimilate a wider context by utilizing a larger field of view (the space of the input vector). The methodology provides an efficient way of balancing correct localization with context absorption without requiring extra computations. Rather than using general convolutions, atrous convolutions are used, which helps to extract more contextual information. The convolutional operation, as described in (Equation 6) is a standard operation that uses a dilation rate of 1.

In contrast, the following equation describes the atrous convolution operation, if the dilation rate is greater than 1.(7)$$\begin{array}{c} \mathrm{FM}\left(x,y,k\right)\sum_{i=1}^{m_{h}}\sum_{j=1}^{m_{w}}\sum_{k=1}^{m_{c}}I_{x+i,y+j,k.r}*K_{i,j,k}\;\;r\geq 2 \end{array}$$ The dilation factor increasedsby ‘2’ with each successive two-dimensional convolution layer. Figure 6 shows how translating a filter’s input image by dilation rates 1, 2, and 4 affects its view on the input image.

This network uses leakyReLU to multiply all negative values by a fixed scalar and transform them into positive ones. To normalize the data, a batch normalization layer is used in the network. The hyperparameters used for the training of the network are shown in Table 1.

### 3.4. Classification Model

Melanoma and benign skin cancers were classified using a novel deep convolutional neural network (N-DCNN) architecture. The N-DCNN for melanoma detection is a carefully crafted network that extracts low-level and high-level skin details by arranging many layers in an orderly fashion. The N-DCNN architecture consists of 11 blocks, as shown in Figure 7.

A deep network is constructed by repeating a few blocks such as 2, 4, and 2 times. A three-channel RGB image is accepted by the network’s first layer, which has dimensions of 128×128. Then, it applies a convolutional operation that slides 8 kernels of size 3×3 over the image. The multiple convolutional layers in several blocks calculate a rich feature set from skin cancer images, which is passed to the next layer, max-pooling. The pooling operation reduces the size of the feature maps by taking the maximum values. In each block, the convolution layer is followed by batch normalization and the leakyReLU activation layer. Normalizing the feature maps from the previous layers is accomplished through batch normalization. To prevent overfitting, it regulates the network’s learning process. In this case, leakyReLU is used since it has a slight slope for negative values instead of a zero slope for standard ReLU. A negative value is converted into a positive value using the leakyReLU function by multiplying it by a scalar value s=0.3:(8)$$leakyReLU=\left\{\begin{array}{c} x\times s,\;x<0\\ x,\;\;\;\;x\geq 0 \end{array}\right.$$ By constructing a deep network of repeating blocks, edge, color, and pattern information about the lesions can be obtained. To interpret the probability of falling into a certain category, the softmax function calculates a confidence score. It is less complex and lightweight than other state-of-the-art networks because it generates fewer learnable parameters (3.3 M) and kernels (3.1 K). A backpropagation algorithm is used to optimize the weights by reducing the loss-concerning gradients. To make small changes in the direction of optimization, a stochastic gradient optimizer updates the network’s weights and biases.(9)$$\theta_{i+1}=\theta_{i}-\alpha ▽L\left(\theta_{i}\right)$$ Iterations are represented by *i*, learning parameters (0.001) are represented by α>0, parameter vectors are described by θ, and the loss function’s gradient is defined by $▽L(\theta_{[}i])$. At each iteration of the algorithm, the gradient is evaluated, and the parameters are updated over a small set of batches. Often, networks become stuck in local minima due to large weight values. The gradient descent algorithm was modified to reduce oscillations using a momentum term γ:(10)$$\theta_{i+1}=\theta_{i}-\alpha ▽L\left(\theta_{i}\right)+\gamma \left(\theta_{i}+\theta_{i+1}\right)$$ The error is calculated using the cross-entropy loss function as follows:(11)$$Loss=\frac{1}{N}\sum_{i=1}^{K}\sum_{j=1}^{N}w_{i}T_{i_{j}}log\left(P_{i_{j}}\right)$$ A network determines the weights *w* by taking into account the number of observations *N* and the number of classes *K*. The hyperparameters utilized by the N-DCNN for training are given in Table 2.

## 4. Results

The N-DCNN was evaluated on a large dataset, ISIC 2020, that is composed of dermoscopic lesion images for the classification of skin lesions. A N-DCNN classification network was trained and evaluated in two ways, as explained in Figure 2: the first using raw image datasets after applying general preprocessing operations such as data normalization and augmentation, and the second using enhanced and segmented images. Network training was conducted on GeForce GTX 1080 Ti hardware with a computation capacity of ‘7.5’. The original dataset contained many duplicates that were not suitable for network training. Thus, to eliminate duplicate samples in a dataset, the algorithm given in Figure 8 was designed. In the case of identical images, when the correlation coefficient was greater than 0.99, one copy of each image was discarded.

The distribution of data after removing duplicates is described in Table 3. It can be seen that the adopted datasets were highly skewed among the two skin cancer types i.e., MEL and BEN. In the network training, data augmentation was applied using three common operations: rotation from −300 to +300, scaling by 0.8 in the X direction, and translating by −5 and +5 in the Y direction. The training set was only subjected to these operations, whereas the validation and test sets were not altered, and their original data distribution was used.

To compare the model’s performance with other state-of-art-methods, several metrics [48,49] were considered, such as precision (PRE), recall (REC), accuracy (ACC), specificity (SPE), F1 score, and learnable parameters.(12)$$ACC=\frac{TP+TN}{TP+FP+TN+FN}$$(13)$$PRE=\frac{TP}{TP+FP}$$(14)$$REC=\frac{TP}{TP+FN}$$(15)$$SPE=\frac{TN}{TN+FP}$$(16)$$F1\text{-}\mathrm{Score}=\frac{2TP}{2TP+FP+FN}$$

The terms TP, FP, TN, and FN represent true positive, false positive, true negative, and false negative, respectively. The ACC measures the percentage of correctly identified samples to the total number of predictions. PRE and REC are significant metrics when assessing the performance of a model, since PRE measures all positive predicted rates, and REC determines the actual positive ratio from all positively identified samples. A metric called SPE measures the model’s ability to identify the TN of each class. A harmonic mean can be calculated by considering FP and FN when computing the F1 score. A value near one indicates the perfect PRE and REC.

The total number of parameters of the N-DCNN was calculated as the sum of all layer’s parameters including the convolutional layer, fully connected layer, and batch normalization layer. The pooling layer parameters were not added as the pooling layer does not contribute parameters. The parameters for each layer were calculated using different formulas.
For the convolution layer, the parameters were calculated asParameters=(Kernel_width×Kernel_height×no._of_kernels_in_the_previous_layer+1)×no._of_kernelsAt the fully connected layer, the parameters are calculated asParameters=(current_layer_neurons,c×previous_layer_neurons,p)+1×cFor the batch normalization layer,Parameters=2×number_of_features

The ISIC validation dataset contained a different proportion of samples from both classes, and hyperparameters were optimized to improve performance on the validation dataset. The trained model with fine-tuned parameters was used to evaluate the test set, which was unseen by the network. The performance of the N-DCNN on the ISIC validation set is shown in Figure 9a by plotting its performance between loss and accuracy over the number of epochs. With an increasing number of iterations per epoch, the accuracy of the network gradually increased. The confusion matrix given in Figure 9b demonstrates that the model achieved TPs of 1293, TNs of 1050, FNs of 108, and FPs of 126 on the ISIC test dataset. On the ISIC datasets, the area under the curve (AUC) for the MEL class was 0.9642, which illustrates the trade-off between sensitivity and specificity achieved by the model. Figure 9c shows the true positive vs. false positive curves.

As shown in Table 4, the N-DCNN performed better than the other state-of-the-art approaches in skin lesion classification. Based on metrics such as ACC, PRE, REC, SPE, and F1 score, the most impressive results are highlighted in bold. For the ISIC 2020 dataset, N-DCNN achieved higher values for all metrics than the other methods. The winners of the ISIC 2020 challenge were determined primarily by their AUC. There were quite a few duplicates in the dataset, and, to the best of our knowledge, no study has tried to eliminate duplicates. In this analysis, the N-DCNN model achieved the highest AUC score among the considered methods, as shown in Table 5.

Furthermore, in Table 6, a comparison between the proposed N-DCNN and some efficient baseline CNN models such as ResNet18, Inceptionv3, and Xception is presented. The comparison was performed between these popular networks on the same dataset. In Table 6, it is evident that the proposed model performed better on the ISIC test set than the other networks in terms of ACC, PRE, and RECm with the fewest learnable parameters, making it lightweight and simple.

The preprocessed and segmented images were used to further evaluate the performance of the classification network. The IA-HR algorithm was applied first to remove hairlines, and contrast was increased using MCACNN before DilatedSkinNet segmentation was used to extract the accurate ROIs from the lesion. With the preprocessed training, validation, and test samples of the ISIC dataset, the N-DCNN model was again trained, validated, and evaluated. On the preprocessed data, the N-DCNN model was trained with the same tuned hyperparameters. Table 7 and Figure 10 show the performance of the N-DCNN on the raw and preprocessed data.

## 5. Discussion

Skin cancer prediction as malignant or benign is accomplished using dermoscopic images of the skin. Dermoscopy is the safest and most time-effective [57] mode of capturing these lesions. There is a great need for computerized diagnostic systems to determine types of skin cancer as the number of cases is continuously growing all over the world [58]. Clinical analysis traditionally analyzes the lesions according to the ABCDE [59] criteria, which is time-consuming, and diagnosis is difficult. Moreover, the lesions’ high inter- and intra-class similarities, morphological complexities, and variations have further motivated the development of an automated classification framework. Various lesion image features such as textures, geometrical aspects, color, and shape characteristics are used to determine skin conditions [41]. The development of an efficient model that can capture the in-depth feature details of a lesion image for accurate classification is required. Thus, in the proposed work, a novel approach named N-DCNN, inspired by convolutional neural networks, was designed for dermoscopic skin cancer classification. The classification model was developed via the careful organization of deep learning layers; further, it was trained and tested on the publicly available ISIC 2020 dataset. The dermoscopic image dataset consists of duplicate images; thus, to overcome bias issues, the duplicate removal method was designed based on a correlation index. Further, a data augmentation method was applied to balance the classes and to avoid overfitting. The presence of hairlines, low contrast, and unwanted background area were considered in this investigation regarding whether they impact classification accuracy. From the statistical results presented in Table 7, it can be seen that the preprocessing operations including the removal of hairlines, increasing contrast, and segmenting lesions significantly improved the performance of the N-DCNN classification network. For each stage of preprocessing, novel approaches were developed such as a hair removal method Section 3.2.1 based on morphological operations, and a contrast enhancement model Section 3.2.2 to regain the lost texture contrast of the lesion during the hair removal process. After this, ACNN, a segmentation deep learning model Section 3.3 based on the atrous dilation concept, is applied to remove unwanted pixels in the background. Through the computation of rich features at multiple layers, thN-DCNN efficiently analyzes very complex lesion patterns. The model gained an ACC of 90.92% using raw data and 93.40% using preprocessed data. Furthermore, the inference time for raw images was 76 s. On the other hand, the preprocessed images required only 17 s for the same test set. The low inference time can be attributed to the fact that the network takes less time to compute lesion features from an image that contains the ROI only rather than processing the whole image. For melanoma classification, the test time on a single image using N-DCNN was quite low at 1.8 s on raw and 1.3 s on preprocessed data. It was found that N-DCNN could improve the accuracy and reduce the architectural and computational complexity. Table 4 demonstrates the classification performance of the proposed N-DCNN model with state-of-the-art classifiers. The method by S. Nasiri in [50] achieved an average accuracy of 75%. M.R.Hassan [51] performed a comparative analysis of various deep learning approaches and listed VGG16 as the best model that achieved 93.18%, but the number of parameters was higher. Brinker [52] designed a CNN model on 12,378 open-source dermoscopic images and obtained a precision rate of 74.10%. The study compared the results of 100 images with 157 dermatologists, and deep learning models outperformed the experts. Another study by Kwasigroch [53] achieved a low accuracy rate of 77%. When compared with these studies, the proposed N-DCNN model showed promising accuracy of 91% on raw images and 93.4% on preprocessed images. Further, these studies have not considered the issues of hairlines, low contrast, and segmenting ROIs. Moreover, the studies [54,55,56] mentioned in Table 5 did not remove the duplicates from the ISIC 2020 dataset. N-DCNN performed well despite the irrelevant visual indicators and imaging artifacts when evaluated on a large and enhanced dermoscopic image dataset.

## 6. Conclusions

In this work, a deep convolutional neural network was designed for melanoma vs. benign classification. An automatic system for detecting melanoma would play a significant role in the early diagnosis of this global health problem. Skin cancer occurrence is quickly increasing and influencing various communities globally. Many have lost their lives because of slow testing processes and the lack of facilities because this cancer was not diagnosed in the early stages. This work proposed a novel N-DCNN model for the binary classification of melanoma and benign, which was designed by organizing different deep learning layers and by hypertuning network parameters. Resizing, oversampling, and augmentation operations were applied to the data before the training process. The experimental results showed that the N-DCNN model outperformed CNN models such as ResNet18, InceptionV3, Xception, and other related methods [50,51,52,53]. The model also achieved the highest AUC compared with the related methods [54,55,56]. As a diagnostic tool, the N-DCNN model demonstrates a high true positive rate and is reliable in predicting the correct lesion category. In addition, other major contributions of this work are the development of an image duplicate elimination method, a hairline removal algorithm, image contrast enhancement, and a segmentation model. The proposed N-DCNN was evaluated using preprocessed datasets by removing hairlines as well as applying image enhancement and segmentation approaches to improve its performance. A significant improvement in N-DCNN performance was obtained with preprocessed data, achieving 93.4% accuracy, illustrating the importance of the preprocessing and segmentation operations.

## Figures and Tables

**Figure 1 sensors-25-00594-f001:**
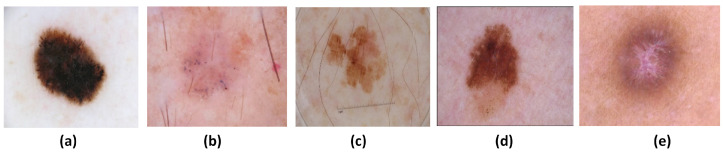
Images of different skin cancer types: (**a**) MEL, (**b**) BCC, (**c**) BKL, (**d**) CMN, (**e**) BEN.

**Figure 2 sensors-25-00594-f002:**
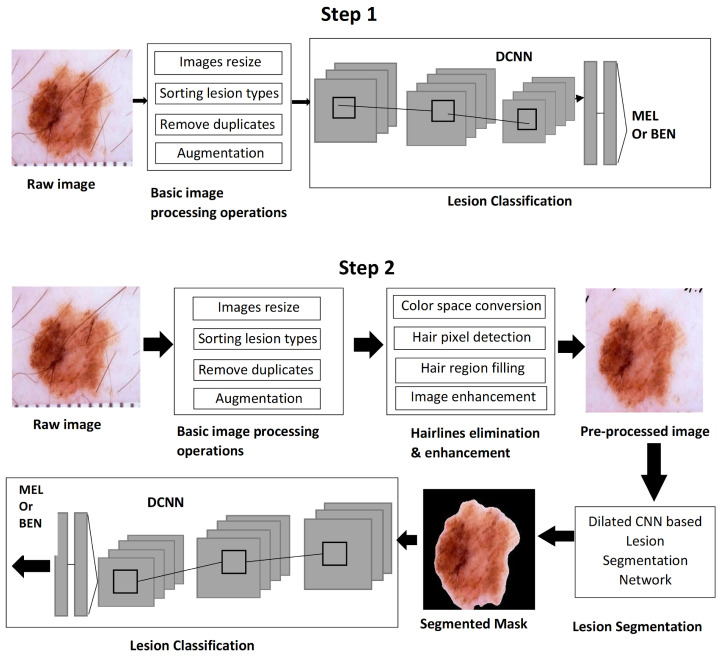
Overall design of the proposed system.

**Figure 3 sensors-25-00594-f003:**
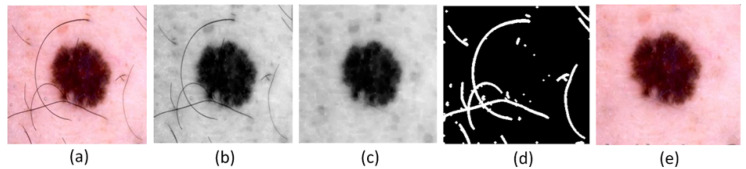
(**a**) Input image; (**b**) grayscale; (**c**) dilation, erosion, and closing operations; (**d**) hair mask; (**e**) noise-free image.

**Figure 4 sensors-25-00594-f004:**
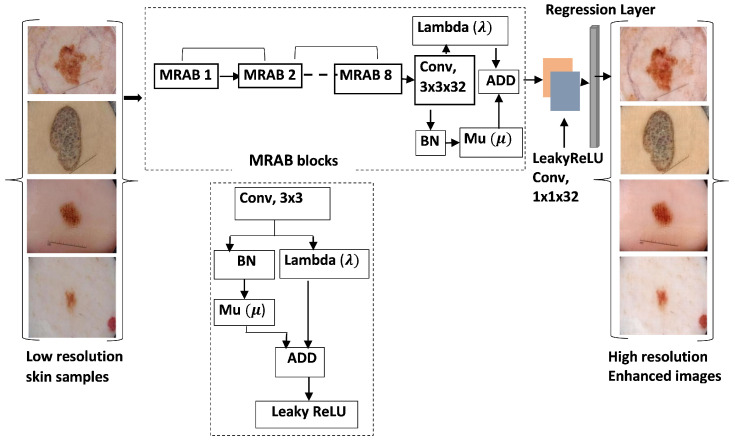
Overview of the MCAN model.

**Figure 5 sensors-25-00594-f005:**
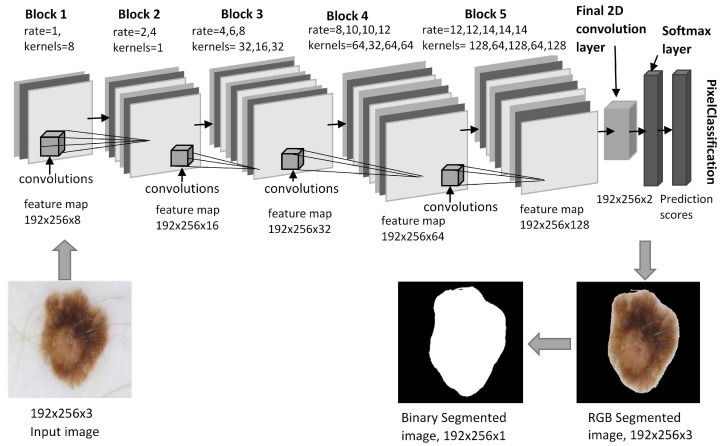
The architecture of the segmentation model (ACNN).

**Figure 6 sensors-25-00594-f006:**
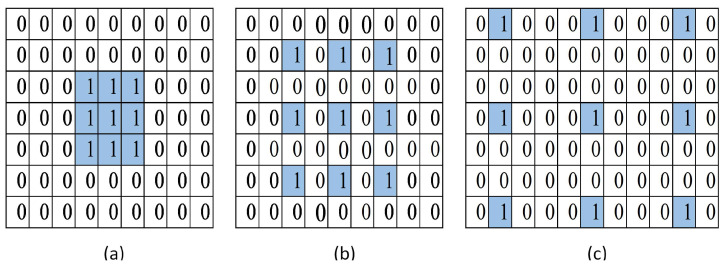
Atrous convolution operations by selecting the blue pixel using the dilation rates. (**a**) rate = 1, (**b**) rate = 2, (**c**) rate = 4.

**Figure 7 sensors-25-00594-f007:**
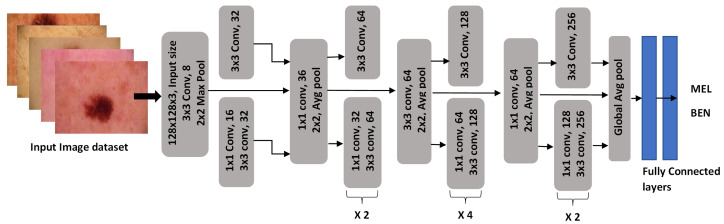
The architecture of a classification network, N-DCNN.

**Figure 8 sensors-25-00594-f008:**
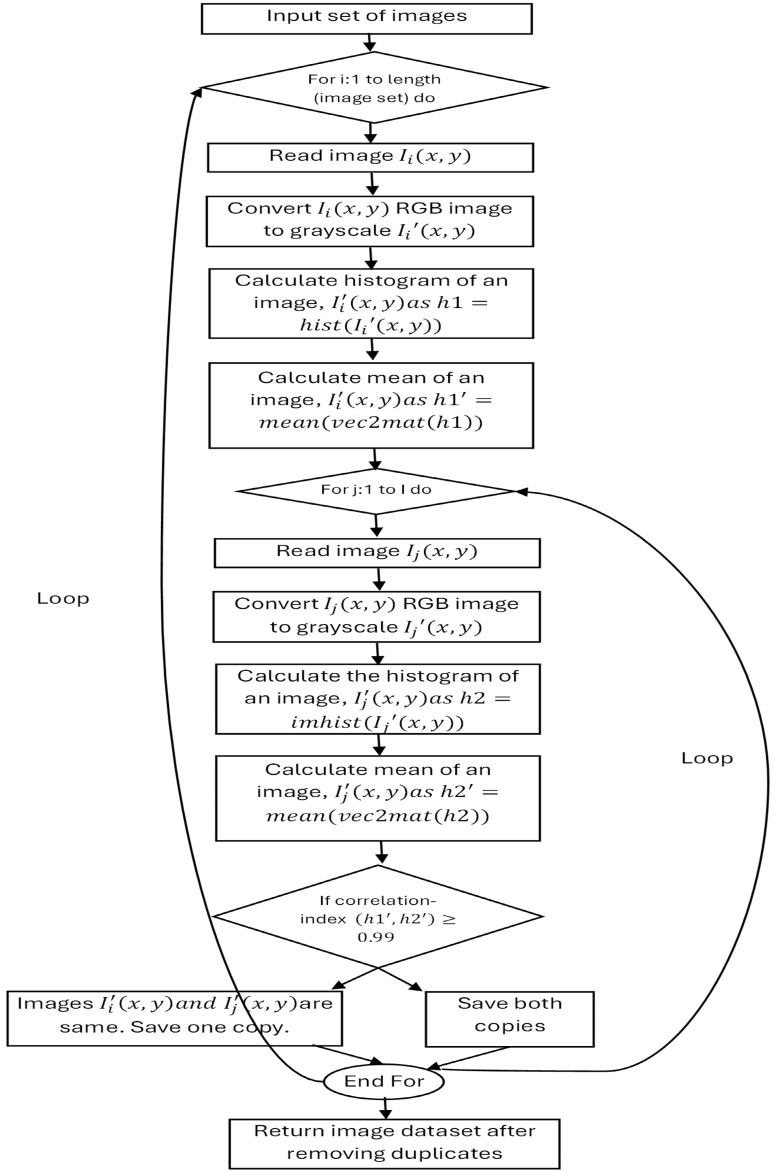
Duplicate image elimination method.

**Figure 9 sensors-25-00594-f009:**
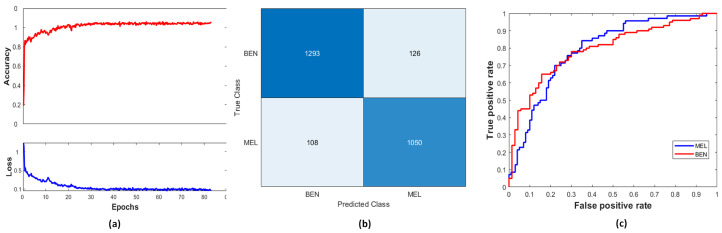
Performance of classification network: (**a**) ACC vs. loss, (**b**) confusion matrix, (**c**) sensitivity vs. specificity curves.

**Figure 10 sensors-25-00594-f010:**
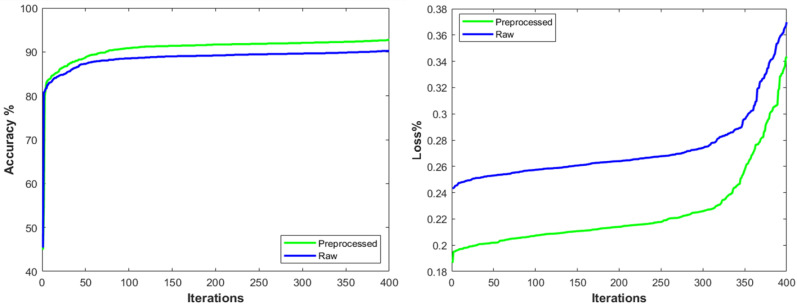
Performance illustration of N-DCNN model in terms of accuracy and loss with preprocessed and raw images.

**Table 1 sensors-25-00594-t001:** The training parameters of the ACNN.

Parameter	Values
Input image size	192×256×3
Batch size	16
Learning parameter, α	0.01
L2 regularization	0.005
Momentum, γ	0.9
Epochs	30
Loss function E(θ)	weighted cross-entropy
Optimiser	SGDM

**Table 2 sensors-25-00594-t002:** Hyperparameters for the training the N-DCNN.

Parameter	Values
Learning algorithm	SGDM
Learning rate	0.001
Mini-batch size	32
Epochs	100
Activation function	Leaky ReLU
Data augmentation	Random oversampling, rotation,
	translation, and scaling
Momentum	0.99
Regularization	0.0005

**Table 3 sensors-25-00594-t003:** Distribution of ISIC data among training, validation, and test sets.

Classes	Training Samples	Augmented Training	Validation Samples	Test Samples	Total Samples
MEL	4052	4965	579	1158	5789
BEN	4965	4965	709	1419	7093
Total	9017	9930	1288	2577	12,882

**Table 4 sensors-25-00594-t004:** Comparison of N-DCNN with other state-of-the-art methods.

Studies	Dataset	No. of Images	ACC%	PRE%	REC%	SPE%	F1 Score%	Parameters (Millions)
S. Nasiri [50]	ISIC	1346	75.00	77.00	73.00	78.00	75.00	―
M.R.Hasan [51]	ISIC	6594	93.18	―	―	―	―	134.2 M
T.J Brinker [52]	ISIC	12,378	―	74.10	87.50	60.00	―	>23 M
Kwasigroch, A. [53]	ISIC	13,600	77.00	―	―	―	―	7.18 M
**Proposed**	ISIC	12,882	**90.92**	**91.12**	**92.29**	**89.29**	**91.70**	**3.32 M**

**Table 5 sensors-25-00594-t005:** N-DCNN performance compared to other related methods based on AUC.

Studies	Dataset	Duplicates Removed	Model	AUC
S. Karki [54]	2020	NO	Ensemble Nets	94.11
Q. Ha [55]	2020	NO	Ensemble Efficient-Net	94.90
M. O’Brien [56]	2020	NO	DNN	59.1
Proposed	2020	Yes	N-DCNN	**96.42**

**Table 6 sensors-25-00594-t006:** Comparison of proposed N-DCNN with baseline CNNs on the ISIC datasets.

Approach	ACC	PRE	REC	SPE	F1 Score	Parameters	Test Time
ResNet18	85.32	84.86	88.54	81.54	86.66	11 M	171 s
Inceptionv3	90.50	92.82	90.52	90.48	91.66	24 M	260 s
Xception	82.7	80.0	79.8	81.9	82.1	22.8 M	202 s
**N-DCNN**	**90.92**	**91.12**	**92.29**	**89.29**	**91.70**	**3.32 M**	**76 s**

**Table 7 sensors-25-00594-t007:** Performance comparison of N-DCNN on the preprocessed data.

Data	ACC	PRE	REC	SPE	F1 Score	Execution Time (s)	Per Image Test Time (s)
Raw data	90.92	92.11	92.29	89.29	91.70	76	1.8
Preprocessed data	**93.40**	**93.45**	**94.51**	**92.08**	**93.98**	17	1.3

Note: Execution time was for the complete test dataset. The best values are marked in bold.

## Data Availability

The data were obtained from a publicly available benchmark dataset: https://www.isic.org/.

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
