# Peer review of "Advanced Deep Learning Models for Melanoma Diagnosis in Computer-Aided Skin Cancer Detection"

_sensors, 2025, doi:10.3390/s25030594_

Round 1
Reviewer 1 Report
Comments and Suggestions for Authors
This study applies deep learning methods to the detection of melanoma. The topic is highly relevant, and the proposed approach shows promise. However, there are several issues that need to be addressed before the paper can be accepted.
1. In the abstract, the authors state that CAD stands for "computer-aided design." However, in the introduction, CAD is also used as an abbreviation for "computer-aided diagnosis." This inconsistency could lead to confusion. The authors should ensure that the abbreviation is used consistently throughout the paper, with the correct definition clearly stated.
2. In the introduction, the authors inconsistently abbreviate the names of different types of skin cancers. For example, "Melanocytic nevus" is abbreviated as NV, omitting the word "Melanocytic," and "Benign keratosis" is abbreviated as BKL, even though the full name does not contain the letter "L." The authors should adopt a consistent and logical rule for abbreviating terms to improve clarity and readability.
3. In the related studies section, the authors mention that CNN-based models such as FCN, SegNet, and U-Net have limitations because pooling layers can discard important image information. However, the proposed N-DCNN model also uses pooling layers. The authors should provide a detailed explanation of how their N-DCNN addresses this issue and ensures that important image information is preserved.
4. In the results section, the authors present Algorithm 1 as a block of code using a 'for' loop. It would be more reader-friendly to present this algorithm as a flowchart instead. A flowchart would make the framework and logic of the algorithm clearer and easier to understand.
5. In Table 4, the authors list the number of parameters for each model. The proposed N-DCNN model, which comprises 11 blocks, reportedly has only 3.32M parameters, whereas other CNN-based models can have as many as 134.2M parameters. The authors should clarify how they calculated the number of parameters for their model and verify the accuracy of this calculation.
6. In Table 6, the authors compare the performance of several models. While it is true that a larger number of parameters does not always indicate better performance, the AlexNet model used for comparison has 18 times more parameters than the proposed N-DCNN. Using such a large model for a binary classification task risks overfitting. The authors should consider replacing AlexNet with a more appropriate model for comparison to ensure a fair and meaningful evaluation.
Author Response
Comment: In the abstract, the authors state that CAD stands for "computer-aided design." However, in the introduction, CAD is also used as an abbreviation for "computer-aided diagnosis." This inconsistency could lead to confusion. The authors should ensure that the abbreviation is used consistently throughout the paper, clearly stating the correct definition.
Response:
Section: Introduction, Lines 52 and 54.
The CAD stands for “Computer-aided diagnosis” and is corrected throughout the manuscript to maintain consistency
Comment : In the introduction, the authors inconsistently abbreviate the names of different types of skin cancers. For example, "Melanocytic nevus" is abbreviated as NV, omitting the word "Melanocytic," and "Benign keratosis" is abbreviated as BKL, even though the full name does not contain the letter "L." The authors should adopt a consistent and logical rule for abbreviating terms to improve clarity and readability.
Response: Section: Introduction, Line 28
The abbreviations for skin cancer types used in the manuscript are corrected as Benign Keratosis-Like (BKL) lesions and Congenital Melanocytic nevus (CMN).
Comment: In the related studies section, the authors mention that CNN-based models such as FCN, SegNet, and U-Net have limitations because pooling layers can discard important image information. However, the proposed N-DCNN model also uses pooling layers. The authors should provide a detailed explanation of how their N-DCNN addresses this issue and ensure that important image information is preserved.
Response:
The limitations of CNN models such as FCN, SegNet, and U-Net, often cited in the literature section of the paper, use multiple pooling layers in their architecture, which degrade image resolution during segmentation tasks. These models, designed for segmentation, aim to reconstruct images at the output while extracting key features through multiple network layers. However, the excessive use of pooling layers reduces the size of feature maps as the network deepens, leading to suboptimal results for segmentation.
We developed an atrous convolutional neural network (ACNN) specifically for segmentation tasks to address this issue. Instead of pooling layers, ACNN utilizes atrous (dilated) convolutions, which preserve the spatial resolution of feature maps while effectively capturing detailed object features. This approach enables the reproduction of high-quality segmented images at the output. In contrast, the proposed N-DCNN model, designed for classification tasks, incorporates pooling layers, as they are integral to down-sampling and enhancing performance in classification networks. In conclusion, pooling layers are more suitable for classification tasks than segmentation tasks.
Comment: In the results section, the authors present Algorithm 1 as a block of code using a 'for' loop. It would be more reader-friendly to present this algorithm as a flowchart instead. A flowchart would make the framework and logic of the algorithm clearer and easier to understand.
Response: Page 11.
The given algorithm 1 in the manuscript has been changed to a flow chart as given below and updated in the manuscript.
Comment: In Table 4, the authors list the number of parameters for each model. The proposed N-DCNN model, which comprises 11 blocks, reportedly has only 3.32M parameters, whereas other CNN-based models can have as many as 134.2M parameters. The authors should clarify how they calculated the number of parameters for their model and verify the accuracy of this calculation.
Response: Page 13, Line 306-318
The total number of parameters of the N-DCNN networks is the sum of all layer’s parameters including the convolutional layer, fully connected layer, and batch normalization layer. The pooling layer parameters are not added as the pooling layer does not contribute parameters. The parameters for each layer are calculated using different formulas.
Comment: In Table 6, the authors compare the performance of several models. While it is true that a larger number of parameters does not always indicate better performance, the AlexNet model used for comparison has 18 times more parameters than the proposed N-DCNN. Using such a large model for a binary classification task risks overfitting. The authors should consider replacing AlexNet with a more appropriate model for comparison to ensure a fair and meaningful evaluation.
Response: Page 14, Table 6.
As advised the AlexNet model has been replaced with Xception neural network for more suitable comparisons. This network has been trained on the same adopted datasets and training settings.

Reviewer 2 Report
Comments and Suggestions for Authors
Dear authors, the following are the requested corrections:
References are not considered in the discussion to identify the contribution of this article and should be expanded, updated and focused on similar work.
The contribution of the work is not identified because throughout the document, and especially in the discussion and conclusions, it is not contrasted with references to similar work. The references should focus on works that include image pre-processing in similar pathologies in order to identify the contribution of this article.
It is important to identify in the database used how many samples correspond to female and how many to male patients, the age ranges and the nationalities, since in the introduction it is mentioned which countries present more melanoma and it is not related to the phenotypes.
The article should be presented in an impersonal way, as there are sections, such as the conclusions, where "we" is mentioned.
The samples correspond to 2056 patients, but there is no mention of the 12 882 skin samples (7093 of which are BEN samples and 5789 of which are 194 MEL samples), whether they correspond to the same stage of the disease, whether they were taken from the patient at the same time as the images corresponding to each of them, or whether they were taken over time.
Best regards.
Author Response
Comment: References are not considered in the discussion to identify the contribution of this article and should be expanded, updated, and focused on similar work.
Response: The Discussion section has been updated and expanded, including the work's contributions and comparisons with similar studies. Moreover, comparisons with similar studies are given in Tables 4, 5, and 6. The contributions of the work are also listed at the end of the Introduction section.
Comment: The contribution of the work is not identified because throughout the document, and especially in the discussion and conclusions, it is not contrasted with references to similar work. The references should focus on works that include image pre-processing in similar pathologies in order to identify this article's contribution.
Response: The discussion and conclusion section has been updated and expanded, including the work's contributions and comparisons with similar studies.
Comment: It is important to identify in the database used how many samples correspond to female and how many to male patients, the age ranges and the nationalities, since in the introduction it is mentioned, which countries present more melanoma, and it is not related to the phenotypes.
Response: The given dataset is retrieved from the benchmark ISIC Challenge 2020 https://api.isic-archive.com/collections/70/, which consists of a total of 33126 lesions from 2056 patients. The lesions were collected from males and females. Skin cancer is more common in men than women. In the given dataset, 15981 lesions are from females and 17,145 belong to males.
Comment: The article should be presented in an impersonal way, as there are sections, such as the conclusions, where "we" is mentioned.
Response: The words pointed such as “we” and other related terms are eliminated.
Comment: The samples correspond to 2056 patients, but there is no mention of the 12 882 skin samples (7093 of which are BEN samples and 5789 of which are 194 MEL samples), whether they correspond to the same stage of the disease, whether they were taken from the patient at the same time as the images corresponding to each of them, or whether they were taken over time.
Response: In the dataset, 16 samples were collected for each 2056 patient to observe changes over time, and they belong to different stages of cancer. Because skin cancer develops over a while, its ABCDE (Asymmetry, Border, Color, Diameter, and Evolution) characteristics change over time. Some patients have developed symptoms of cancer that’s why their mole changes in shape, color, and size. Some patients have not observed changes at the time of the next scan; thus, such images are forming duplicates. In real practice, the patient must observe changes in the suspected skin mole over time.

Reviewer 3 Report
Comments and Suggestions for Authors
This paper describes different techniques to classify the potential melanoma. The presentation, the methodology, experimental results, and discussions are fine. On the other hand, the problem has been worked on many years (e.g. one hair removal technique was developed in 1997). It is better to review the past studies other than [54-56] and compared their performance also. Some specific comments are below.
1. Section 3.2.1, any different types of hairs covering skin lesions?
2. Section 3.3, Any considerations of texture and patch of skin lesions?
3. Table 1, why have chosen these hyperparameter values?
4. Section 3.4, random oversampling may cause overfitting. Any consideration?
Author Response
Comment: Section 3.2.1, any different types of hair covering skin lesions?
Response: For the hair removal process from the images, dark, light, thin, and thick hairlines are considered for removal. The proposed hairline removal method is successful in eliminating hairlines except in rare cases where thick hair is present.
Comment: Section 3.3, Any considerations of texture and patch of skin lesions?
Response: The MCAN model is applied to improve the overall contrast of the dermoscopic image. This method improves color contrast and removes light illuminations to contribute rich feature calculations.
There are no patches extracted in the work instead segmentation model ACNN is applied to extract the region of interest that contains lesion information only and ignores the background.
Comment: Table 1, why have chosen these hyperparameter values?
Response: The hyperparameter mentioned in Table 1 has been selected after executing several experiments on the given datasets. The parameter values are tuned by training the models on the training dataset and observing the model’s performance on the validation dataset. The values offering the best performance have been finalized for the model.
Comment: Section 3.4, Random oversampling may cause overfitting. Any consideration?
Response: In this work, Random oversampling is applied to increase samples in the deficient class i.e., Melanoma. The consideration was the whole dataset was divided into training, validation, and test datasets. During experiments, it has been observed that validation and test datasets have almost similar performances in terms of metrics indicating no overfitting.

Round 2
Reviewer 1 Report
Comments and Suggestions for Authors
The authors have solved the issues I raised in the original manuscript.
Reviewer 2 Report
Comments and Suggestions for Authors
The authors have made the requested changes, so I suggest the manuscript be accepted for publication considering this final version.